



# The Colors of Proxy Noise

Mara Y. McPartland[1], Thomas Münch[1], Andrew M. Dolman[1], Raphaël Hébert[1], Thomas Laepple[1,2]

[1]Alfred-Wegener-Institut Helmholtz-Zentrum für Polar- und Meeresforschung, Research Unit Potsdam, Potsdam, Germany
[2]MARUM—Center for Marine Environmental Sciences, University of Bremen, Bremen, Germany

*Correspondence to*: Mara Y. McPartland (mara.mcpartland@awi.de)

**Abstract.** Uncertainty in paleoclimate time series is inherent to the complex biological and physical processes involved in forming and archiving them in the environment for centuries or longer. The timescale-dependency of this uncertainty is often referred to as "noise" of a particular color based on similarities between the power spectrum of a timeseries and the electromagnetic spectrum of light. For example, "white noise" equally affects all timescales, where "red noise" dominates only on long timescales, similar to longwave red light. In paleoclimate research, the frequency characteristics of proxy noise are often assumed based on first principles rather than estimated directly, which risks either inflating or underestimating error at particular frequencies. Here, we synthesize several studies that use a common method to estimate the spectrum of error in ice core, coral, and tree-ring data. We conceptualize how time-scale dependent noise in proxy time series is created through the archive formation and data processing. Our results suggest that the colors of proxy noise are archive- specific, with white noise dominating in depositional archives such as ice-cores and marine sediment cores, while red noise is likely more common in biological archives such as tree rings and corals. Our aim is to clarify these concepts and provide tools for assigning noise terms in proxy system models, data assimilations, and other experiments.

## 1 Introduction

Paleoclimate proxy records archive past climate information via biophysical or depositional pathways and preserve it in rings, layers or strata. The processes that create these records integrate non-climatic variability alongside the climate signal either during the archiving process, or afterwards as the physical record is modified over time (Evans et al., 2013; Jones 2009; Cook 1987). Recovering paleoclimate information from these archives requires sophisticated data processing and modeling techniques intended to extract climate-related variance from noisy time series (von Storch et al., 2004; Cook & Kairiukstis 1990; Hughes & Ammann 2009; Dee et al., 2016). Recognizing that these methods may be imperfect, the challenge lies in minimizing and rigorously quantifying the impact of non-climatic variance on the signal of past climate variability.

Modification of proxies can either add variance by incorporating spurious or stochastic variations, or remove variance through smoothing across observations (Fig. 1). Technically, we regard a process that adds variance on top of an existing climate signal as a "noise process", whereas the removal of variance constitutes error, but not 'noise' per se. Processes that remove variance are typically deterministic to some extent. For example, two ice-core records with similar physical properties are likely to have



been similarly affected by isotopic diffusion (Whilans & Grootes 1985). It is possible to correct individual records for deterministic errors if the process is well-understood (Shiffelbein 1985; Meko 1981; Dolman et al., 2021a; Shaw et al., 2023).

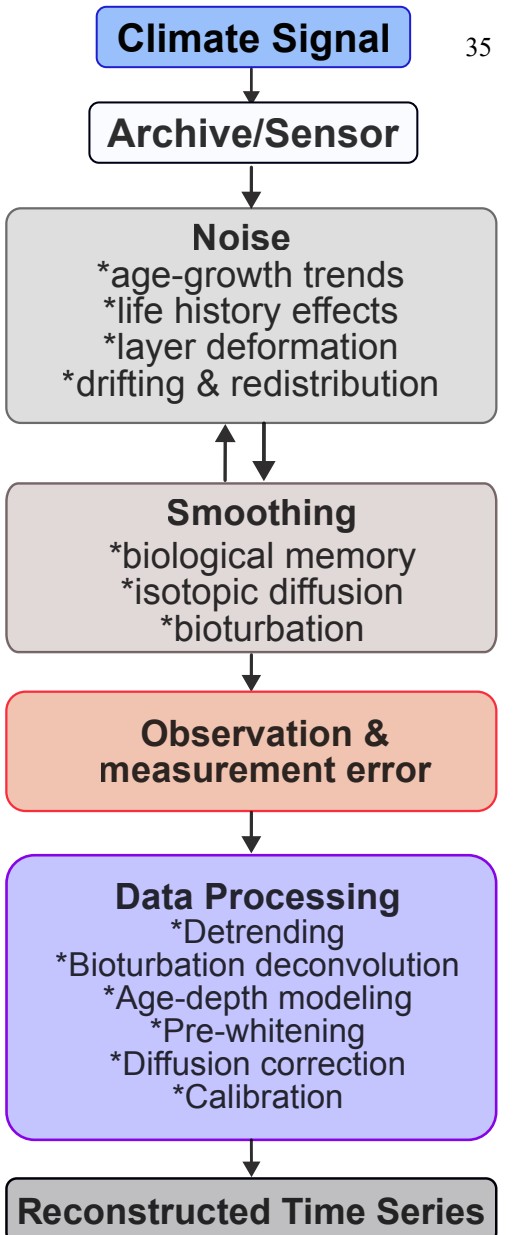

35

**Figure 1: Conceptual diagram showing integration of different types of timescale-dependent proxy errors alongside climate signals via stochastic noise and subtractive smoothing.**

By contrast, additive noise is often independent between sites, generating differences between individual records as well as to the true climate signal. Observation and measurement errors are best represented by uncorrelated noise, unless they represent systematic bias, for example due to a change in the measurement apparatus. Because these types of noise are independent, averaging, or "stacking" individual records reduces noise while retaining the climate signal.

Processes that modify climate signals in proxies result in specific timescale-dependent uncertainties. For example, tree rings contain correlated trends as a result of age-growth effects (Fritts 1976, Speer 2010). Age-growth trends create long-term mismatches between climate and tree-ring data, such that tree-ring timeseries are typically 'detrended' before they are used in reconstructions (Cook & Kairiukstis 1990; Melvin & Briffa 2008; Melvin & Briffa 2014 a,b). Ice cores or sediment records may be modified by physical smoothing processes such as isotopic diffusion or bioturbation within the deposited layers (Johnson et al., 2000; Whillans & Grootes 1985; Hutson 1980; Peng & Broeker 1984). Smoothing dampens the climate signal on fast timescales, becoming less influential on longer timescales (Schiffelbein & Hills 1984; Laepple & Huybers 2013; Münch & Laepple 2018; Bothe et al., 2019).

Proxy error can be characterized in the spectral domain and is often referred to using colors by loose analogy to the frequency spectrum of light (Fig 2). The relationship between power spectral density and frequency is often summarized using a power-law scaling exponent $\beta$ (Box 1) (Vautard & Ghil 1989, Fraedrich & Blender 2003; Hébert et al., 2021). A 'white' noise process implies that the power spectral density is distributed evenly across the frequency space ($\beta=0$), similar to the spectrum of white light. White noise is uncorrelated in time, and is the simplest and most commonly-applied noise model in paleoclimate research (Fisher et al., 1985, Amman & Whal 2007; von Storch et al., 2004; Mann et al., 2005, Lee et al., 2008; Smerdon et al.,



2010). By contrast, processes with relatively more low-frequency variability are termed 'red' noise, by analogy to long-wave red light, represented with a positive slope value (*β>0*), or occasionally 'pink noise' in the specific case when *β=1*. Red or pink noise implies autocorrelated errors that affect low-frequencies at a greater magnitude. (Mann et al., 2007; von Stoch et al., 2009; Smerdon 2012; Zhu et al., 2023). Finally, blue noise refers to processes with relatively higher variability at high frequencies (β<0). Blue noise is characterized by an anti-correlated structure, implying rapidly vanishing effects with increasing timescale (Mann & Rutherford 2002; Mann et al., 2007).

> **Power-law scaling in frequency space**
> - The spectral exponent *β* summarizes the relative contribution of high- and low-frequencies to the total variance.
> - The power spectral density *S(ω)* is assumed to approximately follow a power-law with frequency *ω* such that $S(\omega) \propto \omega^{-\beta}$
> - *β* is typically expressed as the negative slope of a linear regression on a log-log plot of the power spectrum.

**Box 1: Summarizing the timescale-dependency of proxy noise using spectral power-laws.**

The spectra of proxy noise can be either modeled based on mechanistic understanding, or empirically estimated from data. In cases where the physical processes affecting proxies are well-constrained, the power spectrum of the noise can be estimated using parametric models based on biophysical mechanisms (Dee et al., 2016; Dee et al., 2017). The effects of additive noise from measurement error and under-sampling can also be incorporated into mechanistic models of uncertainty (Schiffelbein 1985; Kunz et al., 2020, Dolman et al., 2021b). Proxy errors can also be estimated empirically by comparing time series to instrumental records or climate models (Ault et al., 2013; Franke et al., 2013; Reschke et al., 2019). However, in the former case, noise estimates are restricted to decadal and sub-decadal time scales for which we have instrumental data. The latter case assumes that the medium- and low-frequency behavior of the climate system is correct in the models, and thus that discrepancies reflect proxy noise rather than uncertainty in climate models (Deser et al., 2012; Maher et al., 2020; Laepple et al., 2023).

Alternatively, estimation of noise spectra can be done with relying solely on proxies by exploiting the spatial correlation of climate signals in co-located records. Below, we present noise estimates derived using a simple empirical approach that partitions shared signal from independent variance on all time scales (Münch & Laepple 2018) (Appendix A).We show results from three studies that have applied this approach to ice core (Münch & Laepple 2018), tree ring (McPartland *et al.,* 2024), and coral data (Dolman *et al.,* in prep). The tree-ring and coral data were sourced from global databases compiled by the Past Global Changes (PAGES) initiative (PAGES Consortium 2017; Walter *et al.,*, 2023), and the ice core data represent two large clusters of cores from Antarctica and Greenland (Graf *et al.,* 2002; Weißbach *et al.,*, 2016; Hörhold *et al.,* 2023) (Appendix B). By synthesizing conventional knowledge, evidence from existing literature, and original analysis we aim to deepen a collective understanding of the behaviour of proxy noise and its implications for recovering climate signals from paleo data.

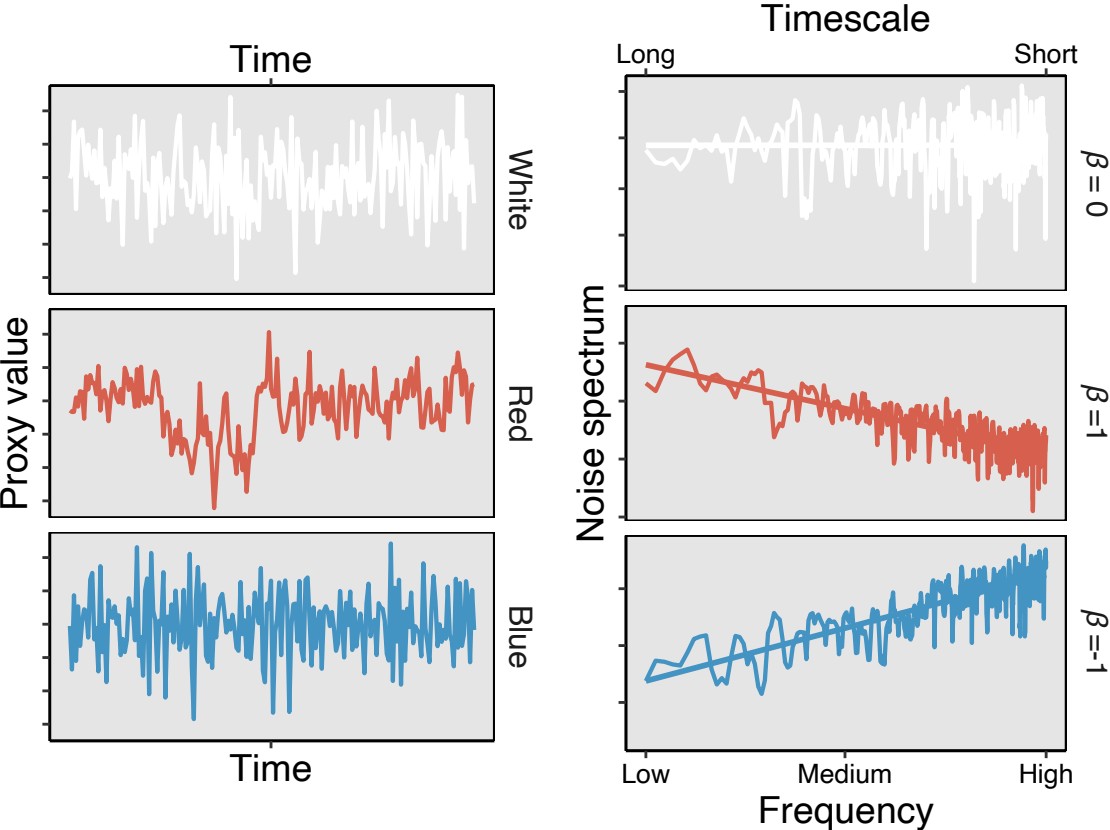

100

**Figure 2: Spectral noise models with an analogy to colored light. Left panels show a simulated time series with the noise spectra shown in the right panels. Top: white noise with no correlation with timescale ($\boldsymbol{\beta}$ = 0). Middle: red noise (sometimes referred to as pink noise) with a positive relationship to timescale ($\boldsymbol{\beta}$ = 1). Bottom: blue noise with a negative relationship to timescale ($\boldsymbol{\beta}$ = -1). Note that $\boldsymbol{\beta}$ values for noise spectra are calculated as the slope of a linear model on a log-log plot, and expressed as $\boldsymbol{\beta}$ = slope*-1,**
105 **following the convention where $\boldsymbol{\beta}$ describes the relationship between power and timescale.**

## 2 The colors of proxy noise

We find that tree rings and corals both exhibit clear red noise spectra with positive scaling exponent $\beta$ values of 0.8 and 0.5 respectively (Fig 3; a, b), such that the power of the noise increases with time scale. As the noise increases more than the climate signal, this leads to a decline of the signal-to-noise ratio with time scale (Fig 3; d, e). Tree-ring and coral records result

110 from the *growth* or *accretion* of layers by an individual organism over time such that biological life history may affect proxy formation. Cambial age impacts both tree-ring width and density, such that detrending to remove juvenile age trends is a near universal practice in dendrochronology (Cook & Kairiukstis 1990). Even after detrending, residual age effects could explain the persistent low-frequency bias in tree-ring records seen here, and observed in other studies (Franke *et al.,* 2013, Ault *et al.,* 2013). Similarly, coral aragonite records might be affected by changes in the biology of individual or descendent polyps over

115 time which may result in a slow drift in the temperature response of the proxy and appear as low-frequency variability, possibly

related to changes in the calcification process (Lough 2004), or persistent baseline shifts in trace element ratios following thermal stress events (D'Olivo & McCulloch 2017; D'Olivo *et al.,* 2019). By extension, red noise might also be a feature of bivalve and sclerosponge chronologies, which contain similar age-growth trends to those found in trees (Jones 1983; Rypel et. al 2008; Hollyman *et al.,* 2018; McCulloch *et al.,* 2024). In general, records composed of repeated measurements made on single long-lived organisms through time are susceptible to ontogenetic effects, the legacies of past disturbances, or slow changes in the behaviour of the sensor.

The stacks of ice cores from both Greenland and Antarctica that we analyzed show a high white noise level where $\beta$ is approximately equal to zero (Fig 3 e, f). As the climate variations become more pronounced on longer time scales, this leads to an increasing signal-to-noise ratio with time. We posit that proxies that are primarily the result of *deposition*, rather than growth or accretion primarily contain white noise. Precipitation intermittency and depositional redistribution in ice cores result in adjacent measurements potentially representing water from different precipitation events (Laepple *et al.,* 2018; Casado *et al.,* 2020; Zuhr *et al.,* 2023). Similarly, in marine sediments where foraminifera or diatoms are deposited from the water column, each sample represents a new set of individuals such that biological effects are uncorrelated between measurements. Therefore, noise in sediment records is also predominantly white with a noise level decreasing as more individuals are measured (Kunz *et al.,* 2020, Dolman *et al.,* 2021). In both ice core and sediment core records, seasonal depositional cycles are much stronger than any interannual or even millennial climate change and the sparse subsampling of the seasonal signal leads to aliasing of independent noise within the signal of annual variation (Kunz *et al.,* 2020).

We identified fewer examples of blue noise processes in the paleoclimate literature. Because its effects diminish quickly with time, blue noise does not introduce error past fast time scales. An example of a true blue noise process is the infilling of troughs on ice sheets as wind redistributes snow causing blue noise in noise in annual layer thickness records from ice-cores (Fisher *et al.,* 1985). Blue noise models have occasionally been used in proxy system models to account for a variety of potential types of error affecting high-frequencies (Mann *et al.,* 2007; Mann & Rutherford 2002).

Like blue noise, smoothing processes predominantly affect high frequencies and become less significant with timescale. Biological memory in trees, diffusion in ice cores, and bioturbation in sediments are all examples of smoothing processes that lead to correlated errors between the climate and the proxy signal which, in theory, can be accounted for using deterministic modeling (Matalas *et al.,* 1962; Berger *et al.,* 1977; Meko 1981; Ruddiman *et al.,* 1980; Whillans and Grootes, 1985). Given such a model, the smoothing effect can be reversed, as applied in our example to ice core data to reverse the effects of diffusion (see Appendix A). If the smoothing process affects the climate signal and the proxy noise equally during deposition or accretion, the signal-to-noise ratio (SNR) is unbiased at all timescales, regardless of whether or not a correction for the smoothing effect is applicable, as is the case for diffusion in ice cores. However, when noise is introduced after smoothing (e.g. measurement noise), the attenuated climate signal on the high-frequency side will be masked by a relatively stronger



noise level , biasing the SNR spectrum downwards toward high frequencies. In any case, knowledge about and accounting for smoothing processes in paleoclimate time series is critical for evaluating the short-term effects of climate forcing events such as volcanic eruptions (Esper *et al.,* 2015; Zhang *et al.,* 2015; Lücke *et al.,* 2015), but is potentially less critical for reconstructing low-frequency variations in climate.

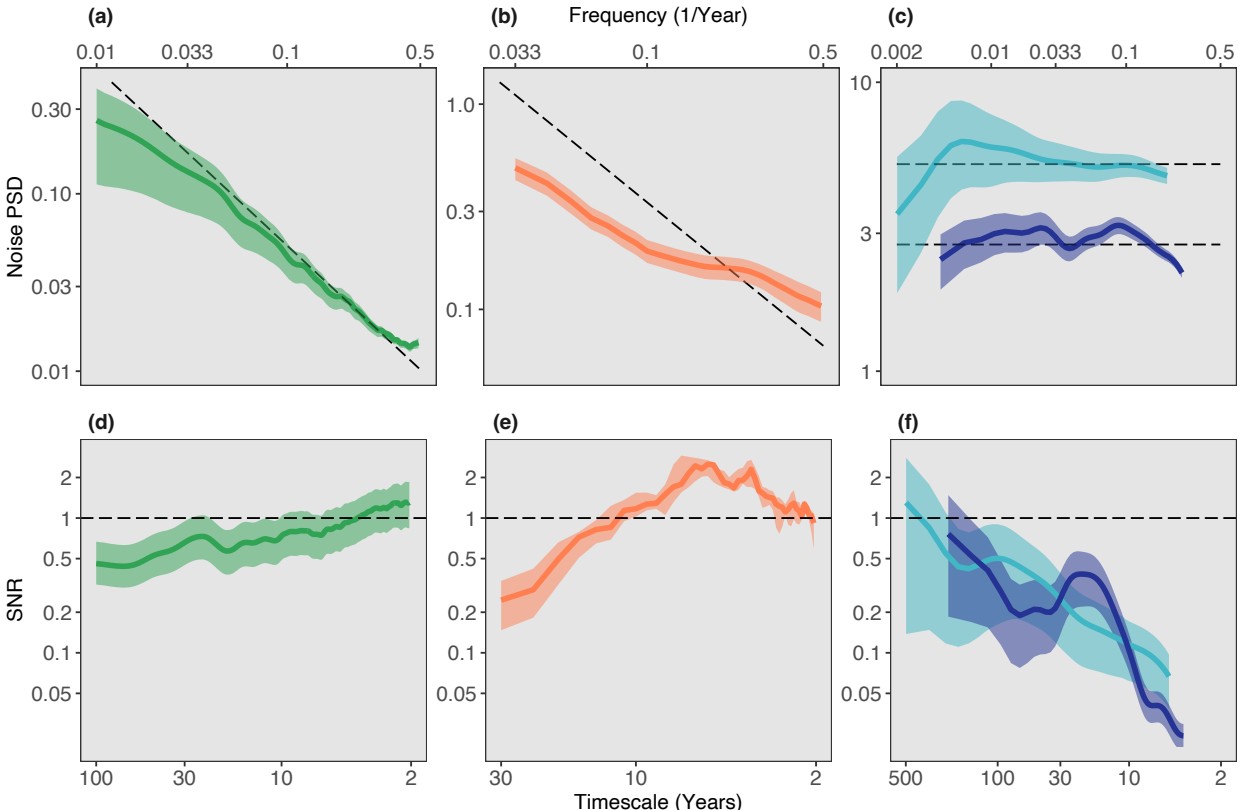

**Figure 3:** **Estimates of proxy noise spectra (a, b, c) and timescale-dependent signal-to-noise ratios (d, e, f). Top: (a) Mean noise spectra for tree-ring width and density records from northern hemisphere tree-ring records, (b) Mean noise spectra for tropical coral $\delta^{18}O$ and strontium/calcium (Sr/Ca) ratios, (c) Noise spectra for ice core $\delta^{18}O$ from Dronning Maud Land (light blue) in Antarctica and the North Greenland Traverse (dark blue). Dashed lines represent an idealized spectral power-law with a slope $\beta =$ 1 for proxies containing predominantly red noise (i.e. tree rings and corals), and with $\beta = 0$ for proxies (i.e. ice cores) containing**
**predominantly white noise. Bottom: Timescale-dependent signal-to-noise ratios (SNR) for (d) tree rings, (e) corals, and (f) ice cores. Dashed lines represent an SNR of 1. Confidence intervals on all spectra represent the 10th and 90th percentiles from a parametric bootstrapping estimation method. Detailed methods for estimating proxy noise and SNR values can be found in McPartland et al., (2024) (tree rings), Münch et al., (2018) (ice cores) and Dolman et al., (under consideration) (corals).**

## 3 Implications

The spectrum of temperature on local to global scales is generally accepted to be red (Huybers & Curry 2006; Cheung *et al.,* 2017; Hasselmann *et al.,*, 1976). For proxies with predominantly white-noise spectra such as ice cores and sediments, this implies that the power spectral density of the climate signal relative to the noise, the signal-to-noise ratio (SNR), increases

with timescale. This explains why ice cores are faithful recorders of millennial climate variability (e.g. EPICA, 2006), while they fail in many regions to reconstruct interannual to decadal changes (Stenni *et al.,* 2017). By contrast, in proxies that contain

red noise, the SNR will rise more slowly or even decline with timescale if the power of the noise rises more steeply than the signal, as we demonstrate in tree rings and corals. These proxies are better recorders of fast time-scale variability where the ratio of signal to noise is highest. For example, corals are faithful recorders of interannual variability and can deliver unique information on tropical climate dynamics such as the El Niño Southern Oscillation (ENSO) (Fig. 3), but have challenges reconstructing multidecadal trends (Scott *et al.,* 2010).


The color of the noise thus determines at which timescales a robust climate signal can be reconstructed. Information about proxy noise can be used to guide future study design (i.e. what proxies can be used to answer a climatic hypothesis) and to optimize the sampling and measuring design (i.e. how many cores are needed; what is the optimal sampling resolution to minimize noise). It can also be used to estimate time scale-dependent uncertainty in climate reconstructions. Error from proxies

with white noise spectra is reduced by averaging in time so in reconstructions that draw on records with white noise spectra, error should be reduced with the addition of more records. In the case of red noise that mimics the spectrum of the climate, uncertainty depends on the slope of the noise relative to that of the climate. If the slope of the noise is steeper than that of the climate, even with averaging in time the error will still overwhelm the signal on the longest timescales. The shape of proxy noise therefore influences the time scales at which estimates of past climate are more or less certain. If unaccounted for, colored

noise can be misinterpreted as past climate variability.

Colored noise models such as those described here are appropriate for use in research activities where the behavior of proxy noise is often assumed rather than estimated. For example, the use of empirical, proxy-specific noise models in pseudoproxy experiments will improve their use in evaluating climate model performance, particularly on long time scales (Jones *et al.,* 2006, Dee *et al.,* 2016; Smerdon *et al.,* 2020). Climate field reconstructions and data assimilation methods often assume white

noise, which risks misconstruing noise as signal, potentially leading to biased results. In climate field reconstructions and data assimilation frameworks proxy-specific noise models could be used to improve the representation of spatio-temporal modes of past climate variability.

**Conclusion**

Here, we present an overview of how colored noise is created and can be represented in different types of paleo archives. By

synthesizing the results of multiple recent studies, we show the distinct nature of noise and signals across archives and discuss how colored noise should be conceptualized in paleoclimate data. These noise models, or models derived using similar methods, can be implemented within paleoclimate research as a way to account for the range of unique biological and physical processes affecting proxies.



**Appendix**

**Appendix A: Estimating the spectrum of noise**

We apply the method of Münch *et al.,* (2018) of combining clustered proxy records into regional stacks and analyzing their variance in the frequency domain. It builds on the assumption that the proxy signal is a function of four main components: the climate signal, additive noise that arises during the proxy creation and archiving stages, measurement noise, and any smoothing processes that act during archiving but not on the measurement noise; i.e.


$$\mathcal{P} = f\left(\mathcal{C}, \mathcal{N}, \Sigma; \mathcal{G}\right) = \mathcal{G}\left(\mathcal{C} + \mathcal{N}\right) + \Sigma$$

where *P*, *C*, *N*, and *Σ* stand for the power spectral densities of the proxy signal, the climate signal, the proxy noise, and the measurement noise, respectively, and where *G* is a transfer function that describes a specific smoothing process such as

biological memory, diffusion, or bioturbation.

Given a regional cluster of *n* proxy records with a similar climate between sites, the mean power spectrum, *M,* averaged across all individual records' spectra, will yield a precise estimate of the proxy spectrum *P*. By contrast, the power spectrum, *S*, of the stacked record from averaging all records in the time domain, will also contain the full climate signal, but with the noise

proportions reduced by a factor of *n*. By combining both quantities one can derive expressions for the climate and noise spectra (Münch and Laepple, 2018),

$$\mathcal{C} = \frac{n}{n-1}\mathcal{G}^{-1}\left(\mathcal{S} - \mathcal{M}/n\right); \quad \mathcal{N} = \frac{n}{n-1}\mathcal{G}^{-1}\left(\mathcal{M} - \mathcal{S} - \frac{n-1}{n}\Sigma\right)$$


with the ratio of *C:N* yielding the frequency-resolved signal-to-noise ratio (SNR). A common smoothing process equally biases the signal and the noise spectrum, if not corrected for by means of the inverse transfer function $G^{-1}$, and hence its effect cancels out in the SNR spectrum. We note that time uncertainty between individual proxy records can be another source of smoothing in the stacked record, but it is less straightforward to include into our methodology (Münch and Laepple, 2018) and is neglected

here.

**Appendix B: Data**

**B.1 Tree Rings**

For the tree-ring data we analyzed the tree-ring records contained within the Past Global Changes 2k (PAGES2k) database, a
large database compiled to reconstruct global temperature variations during the last two millennia. This network of 647 unique
paleoclimate records from around the globe includes 450 tree-ring timeseries, of which we used 421 records of tree-ring width
and density located across the Northern hemisphere (PAGES 2013, 2017; Neukom *et al.,*, 2019). Spatial clusters were defined
using 250-kilometer radii, such that no two sites were more than 500 kilometers apart. Tree-ring width and density records
were clustered separately. This resulted in 186 clusters of sites. More information om the analysis of the PAGES tree-ring
database is available in McPartland et al (2024).

**B.2 Corals**

We used the coral records contained within the PAGES Coral Hydro 2k database to obtain coral SNR estimates (Walter *et al.,*
2023). The Coral Hydro2k database contains 54 oxygen ($\delta^{18}O$) and strontium calcium (Sr/Ca) records from the global tropics.
The database was compiled to reconstruct sea surface temperature and ocean hydroclimate variability for the past two centuries.
Due to fewer records, 1000 km spatial clusters were used, resulting in 64 clusters. $\delta^{18}O$ and Sr/Ca records were clustered
separately and the results are averaged. More information on the coral data curation is contained in Dolman *et al., under
consideration.*

**B.3 Ice Cores**

As an example for ice-core derived temperature proxies, we use stable isotope records from the Dronning Maud Land region
in Antarctica ("DML data" in the following; Graf *et al.,* 2002) and from central-north Greenland ("NGT data" in the following;
Weißbach *et al.,*, 2016, Hörhold *et al.,*, 2023).

The DML data consist of 15 records, 12 of which cover the time period from 1800 to 1998 CE and 3 records cover 1000–1998
CE. We combine both datasets by using the individual spectral results (Münch and Laepple, 2018) of the shorter records on
time scales below decadal and of the longer records on the supra-decadal time scales. We apply the diffusion correction as in
Münch and Laepple (2018) but do not use their time-uncertainty correction.

The NGT data comprise 14 cores covering the time span from 1505 to 1979 CE, including original records from the North
Greenland Traverse published in Weißbach et al (2016) as well as the extended NGT records from exploiting new drillings as
presented in Hörhold *et al.,* (2023). The corresponding NGT spectra shown in Hörhold *et al.,* (2023) were not diffusion-
corrected; here, to be able to compare the NGT spectra to those from the DML data, we apply a diffusion correction to the
NGT spectra following the method given in Münch and Laepple (2018) with diffusion length estimates calculated as described

in Hörhold *et al.,* (2023). Note that the SNR spectrum shown in Hörhold *et al.,* (2023) used the ratio of the integrated signal and noise spectra, which is related to the correlation with the climate signal (Münch and Laepple, 2018), whereas here we
show the direct ratio of the spectra.

**Data Availability**

This work represents a synthesis of multiple independent research projects. The data needed to reproduce the tree-ring and coral data are publicly available through the NOAA National Centers for Environmental Information (Emile-Geay *et al.,*, 2017; Walter *et al.,* 2023). The original Antarctic ice core isotope data are archived at the PANGAEA database (Graf *et al.,*,
2002) as well as the Greenland data except for core NGRIP whose data is available from the Centre for Ice and Climate of Copenhagen University (Weißbach *et al.,*, 2016; Hörhold *et al.,*, 2023). PANGAEA is hosted by the Alfred Wegener Institute Helmholtz Centre for Polar and Marine Research (AWI), Bremerhaven and the Center for Marine Environmental Sciences (MARUM), Bremen, Germany.

**Code Availability**

The general software to conduct the separation of signal and noise in the spectral domain and to perform the signal-to-noise ratio analysis is available as the R package proxysnr from the open research data repository Zenodo (Münch, 2018). Additionally, specific code to reproduce the tree-ring, coral, and ice-core analyses, respectively, are also available via Zenodo (McPartland 2024, Dolman 2024; Münch 2024).

**Author Contributions**

MYM wrote the manuscript, created figures, and contributed the analysis of tree-ring data. TM developed the signal-to-noise ratio analysis, contributed the analysis of the ice core data, and helped write and edit the manuscript. AMD contributed analysis of coral data and the simulations of colored noise spectra, and helped write and edit the manuscript. RH helped write and edit the manuscript. TL developed the signal-to-noise ratio analysis, helped to write and edit the manuscript, and supervised analysis of all proxy data.

**Competing Interests**

The authors declare that they have no conflicts of interest.





## Acknowledgements

This is a contribution to the SPACE ERC project; this project has received funding from the European Research Council (ERC) under the European Union's Horizon 2020 research and innovation program (Grant Agreement 716092). A. Dolman was
supported by the Deutsche Forschungsgemeinschaft (DFG, German Research Foundation) – Project number 468685498 (A.M.D.) – SPP 2299/Project number 441832482. T. Münch was supported by the Informationsinfrastrukturen Grant of the Helmholtz Association as part of the DataHub of the Research Field Earth and Environment. We acknowledge support by the Open Access Publication Funds of Alfred-Wegener-Institut Helmholtz-Zentrum für Polar- und Meeresforschung. The work profited from discussions at the Climate Variability Across Scales (CVAS) working group of the Past Global Changes
(PAGES) program. We acknowledge the contributions of Nora Hirsch, Jannis Viola and Vanessa Skiba, along with members of the AWI-Earth System Diagnostics Group.

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
