# Peer review of "The Colors of Proxy Noise"

_Climate of the Past, 2024_

## Author Response (AR1)

Summary response to all three reviewers & editor (Author responses in italics)

Authors: We thank the editor and three reviewers for taking time to read and comment on our manuscript. We have made every effort to respond to all comments and clarify the aims of our project. We believe that our edits have improved readability and made our argument more persuasive. We have also incorporated more references to the published literature on proxy system modeling, the main other source of information on proxy noise color. We have also expanded our Extended Data section, adding two new appendices with additional figures. The first presents a more comprehensive result that includes the uncorrected and signal spectra, in addition to the noise and signal-to-noise ratio, and demonstrates the sample density at each timestep. The second deals with the effects of time uncertainty on the noise spectrum, for which we rely on data simulated using a range of potential band-counting errors in corals.

Ultimately, we have determined to keep the main structure of our paper, with a review of the literature followed by a synthesis of three results from other published studies, as it was in our initial submission. As we argued in the open discussion phase of the review process, our aim was to write an accessible article on proxy noise characteristics that could guide research in a variety of current research areas within paleoclimatology.

Response to editor comment "I appreciate that the authors have provided a roadmap to how the plan to revise. Personally, I feel that the manuscript may benefit from a brief introduction of time series color in general. While the authors state that sediment core noise is generally white, the time series itself is typically red. I think this could benefit the general reader, making the manuscript more accessible to a larger audience."

Authors: Thank you for this suggestion. We have modified the section where we describe colored noise to be more generally about time series color, and have added in a description of how colored noise interacts with time series color by either 'reddening' or 'whitening' the spectrum of the original time series (Lines 63-67).

Response to Anonymous Referee #1

Summary: The manuscript discusses the spectral characteristics of non-climatic variability in climate proxies, so-called 'proxy noise'. The manuscript formalizes much of the empirical knowledge about the effect of noise on climate reconstructions based on tree-rings, corals and ice-cores.

Recommendation: The information presented in the manuscript is not completely new. It is scattered among the different proxy communities and usually impolitely assumed when reconstructing uncertainty bounds in climate reconstructions. Nevertheless, I found the manuscript interesting and worth publishing, as it presents more formal

characterizations and definitions of noise and discusses the effects that, in general, proxy retrieval and post-processing methods have on the final proxy time series. I think this can be a nice contribution to the different proxy communities.

The manuscript does not touch upon one important source of 'noise', namely dating uncertainty, which can be substantial for some proxies and negligible for others (e.g. tree-rings). One appendix briefly indicates that it lies outside the scope of the study, but I think this should be mentioned in the main text and possibly also briefly discussed.

My recommendation is, therefore, that the manuscript needs a few minor revisions. The author may want to consider my suggestions below.

1) Dating uncertainty is not considered in this manuscript, but many readers would precisely expect to read what the effects of this uncertainty could be. Could the authors include a brief discussion, perhaps without formal treatment, pointing to a follow-up study?

Authors: We thank the reviewer for this suggestion. We have added in a new section in the discussion on time uncertainty, how it affects the signal-to-noise ratio of the three proxies we investigate here (Lines 216-234) with references to the existing literature. We also have added a new section to the appendix (Appendix D) that explores the effects of dating errors (at rates of once per 10 years, 50 years, and 100 years) on simulated proxy signal-to-noise ratios. This includes a new figure (Fig. A2).

2) 'whereas the removal of variance constitutes error, but not 'noise' per se'

I struggled to understand this sentence. I could get the meaning in the end, but it needs a clearer phrasing.

Authors: This sentence has been changed (Line 32)

3) ' represented with a positive slope value ( $\beta$ >0)'

I think it is clear here that the authors are referring to the function of power density as a function of frequency spectral\_power(f). However, some readers may also think in terms of period instead of frequency. The function spectral\_density(T) as a function of period is not just a variable substitution since there is an additional factor:

$$P(f)df = P'(T)dT$$

Since dT and df are not linearly related, there is an additional multiplicative factor that depends on T as well. Perhaps the authors may want to include a warning for those readers. This would also include Figure 2 and its caption.

Authors: If we understand correctly, the reviewer is commenting on the inverse relationship between frequency and timescale which can be confusing to those unfamiliar with spectral analysis. In response we have added the following text (Line 78-82), and added reference to this in the caption of Fig. 2.

"The exponent β represents the relationship between frequency (or time period) and power spectral density, which appears as a linear relationship on a log-transformed plot. By convention, the exponent is defined as the negative of the relationship with frequency such that a positive exponent actually represents increasing variance/power spectral density with timescale."

4) 'Alternatively, estimation of noise spectra can be done with relying solely on proxies by'

by relying

Authors: We corrected this error.

5) 'on ice sheets as wind redistributes snow causing blue noise in noise in annual layer thickness records from ice-cores #

blue noise in noise sounds harder to understand than it should

Authors: We removed the repeated word.

Response to Anonymous Referee #2

Review of the manuscript "The Colors of Proxy Noise" by Mara McPartland and colleagues submitted for publication in *Climate of the Past*

General:

The authors present set out to conceptualize the colors of proxy noise for different proxy archives. For this they use results that are already published or soon will be, in conjunction with already published data sets. They conclude that their models can be used by the community to account for the range of specific biological and physical processes influencing the proxy system.

In general the manuscript is prepared in a very superficial and simplistic style. The abstract does not present any specific hypotheses or questions that are addressed in

the body of the manuscript., i.e. why their study is important and which are the specific new results found?

Authors: We thank referee #2 for taking the time to review our work, and appreciate their willingness to read a new draft.

We understand the referee's concerns, and regret that we did not make the intention and goals of the manuscript clear enough. Our intention is to provide a synthesis of recent insights into the correlation structure of proxy noise in three different proxy archives. The reason that the manuscript is short and doesn't contain a traditional methods section is because the methods used to derive these estimates are available in the referenced specialized publications.

We have opted to leave the structure of the manuscript as it was originally, but we have tried to be very clear that this is a synthesis at multiple points throughout the paper, including the abstract (Lines 15-16), and at the end of the introduction (Lines 121-128), in the discussion (Line 137) and in the conclusion (Line 274).

In our revision, we have made the effort to situate our work alongside the literature from the proxy system modeling community, and discuss the value and limitations of different approaches to noise estimation including ours. We have also expanded the main chapter entitled "The colors of proxy noise" to include additional references and a more extensive discussion on the origins of noise in different records, for example a section on detrending in tree rings (Lines 154-116), and biological processes in corals (Lines 172-177).

The main part also does not include any proper scientific setup with a clear description of methods and a concise description of results, justifying the added value of the manuscript. Therefore the content related to new results is in my opinion not enough to be published in a manuscript in CP, when large parts of the conceptually ideas are already published by other studies in the recent decade (c.f. Smerdon, 2012, Evans et al, 2013, Dee et al., 2017). What is also unclear to me is why the appendix is 30% of the overall length of the (very short) manuscript and does not form a regular method section??

Authors: We respectfully disagree with the reviewer's assertion that a large part of the conceptual ideas presented in our work have been previously published. In the open discussion phase of the review, we argued this and described the differences between our findings and existing literature: <a href="https://cp.copernicus.org/preprints/cp-2024-73/">https://cp.copernicus.org/preprints/cp-2024-73/</a>

In our revision we have made the effort to review these other studies and argue that these approaches are complementary to each other. We have added a new section to the introduction that reviews the literature on proxy system modeling (98-111) and a section in the discussion on applications for colored noise models in pseudoproxy experimentation, data assimilation and climate field reconstructions (Lines 261-271).

With respect to the appendix being long and not forming a regular methods section, this was also intentional as these methods have mostly been published elsewhere, with the exception of the North Greenland traverse result. Because this is a new analysis, we need to maintain the description of this ice core data and analysis in the appendix.

I suggest to reject the manuscript in its present form and the authors should completely re-think their setup and present substantially new results in a revised version.

**Specific:**

**Abstract/Title:**

The Title is very unspecific. Authors should be more specific. Actually it is more promised than the article really holds. The abstract also does not contain substantially new results compared to studies cited mentioned or used in the manuscript.

Authors: The title expresses the aim of the paper to review the structure or 'color' of noise in proxy-records. We have opted to leave the title as-is.

**Introduction:**

The introduction is a summary of results achieved so far in the context of proxy forward and proxy system modelling. No clear question is formulated. As much as i can understand the study just contains a summary of results presented already elsewhere (c.f. "We show results from three studies that have applied this approach to ice core (Münch & Laepple 2018), tree ring (McPartland et al., 2024), and coral data (Dolman et al., in prep)." on page 3.

Authors: In our revision, we have made the effort to place our work within the context of the existing literature on proxy system modeling, and argue that empirical and mechanistic modeling approaches are complementary steps in proxy noise estimation. We also outline the limitations of using solely forward models to estimate proxy noise (Lines 98-111).

In addition, also the information contained in Text boxes and Fig 1 and 2 is published elsewhere and does not present any new result.

Authors: The schematic diagram (Fig 1) is modeled off of a similar one from Evans et al (2013), which presents a simple and straightforward overview of paleoclimate proxy

formation. A modified version of the same diagram appears in Dee et al. (2015 and 2018). Fig 2 is certainly not 'new' but is a visualization of colored noise spectra - that in our view is useful for the readers not familiar with thinking in the time and frequency domain. The text box provides an overview of power-law scaling, which as this is intended for a general audience, is intended to provide a straightforward definition for a non-specialist. We have opted to leave these the new manuscript.

**The colors of proxy noise**

The (supposedly) main chapter also contains a loose collection of (qualitative) information that is presented without any context. No methods section is presented, nor a concrete formulation of an hypothesis that should be addressed in the study. Again, the concept authors motivate is already published in detail in former studies.

Authors: We have expanded on this section to draw in significantly more background on each proxy. We have included a thorough overview, for example, of the pitfalls associated with tree-ring standardization and the potential for these to introduce bias (Lines 154-161). We also discuss the effects of biological memory on the noise spectrum (trees) (Lines 164-170), and expand our discussion of non-linear responses to temperature (corals) using additional sources from the literature (Lines 172-1180). We use these to support Fig. 3, which shows the noise spectra derived using our empirical method, described in detail in the cited publications and in the Appendix.

**Implications/Conclusions**

The conclusions contain a vague summary without any concrete answer to a previously formulated question, without any context to studies and literature published so far in the field of research.

Authors: We have expanded the implications section by adding a new paragraph on implications for paleoclimate research activities with reference to the literature on psuedoproxy experimentation and data assimilation (261-271), and have added an additional few sentences to the conclusion about our project's goals (lines 275-278)

**Response to Anonymous Referee #3**

The authors discuss the characteristics of the noise spectrum in climate proxies—tree rings, corals, and ice cores—and argue that the spectral characteristics of noise differ depending on whether the originating materials are formed through biological growth or deposition. The topic itself is important for paleoclimate research, but the characteristics and limitations of these proxies have long been discussed already. As the paper lacks

both a quantitative evaluation of the uncertainty of the proxies across the frequency range and an in-depth assessment of the cause of the noise characteristics, I feel that the addition of new insights is limited. I therefore suggest a major revision. The following are the detailed comments.

Authors: We thank Referee #3 for taking the time to review our work. We respectfully disagree that there are no new insights to be gained from new work on the topic of the spectral characteristics of proxy noise, which we argued in the open discussion phase of the review process, and can be viewed on the CP website.

In our revision, we have made every effort to be clear about the value of these estimates, and to present our findings within the context of papers on proxy noise estimation from proxy system modeling, and have added more comprehensive literature review to our discussion of the origins of non-climatic variance across proxy types. We have also added new text on time uncertainty (Lines 216-226) and it's impacts on the signal-to-noise ratio, with a new figure in the Appendix (Fig D1) that shows the effects of time uncertainty on simulated proxy data.

**1. About data**

It is important to indicate the details of the data used in this study, including the lengths and periods, to assess the validity of the methods and the limitations. Perhaps the authors could plot the time series used in this study in appendix?

Authors: Thank you for this suggestion. In response to this and some of the criticisms below, we have added a new section to the appendix (Appendix C) where we add in an additional figure that should address several of these comments. The figure shows several additional aspects which were not included in the original publication.

- 1) We have added a density plot for each analysis which shows the time periods covered by each cluster in the analysis (Fig. C1; b,d,f,h). While it was not really feasible to include time series (450 time series were included in the tree ring analysis, for instance) this shows the sample density at each frequency. What is visible is where the sample density drops off and how this limits the recoverable SNR to ~30 years for corals and ~100 years for trees.
- 2) This figure includes the 'signal' curve which is used to derive the SNR, and the uncorrected 'proxy' spectrum which represents the integrated signal and noise.

For the comparison of the behavior of proxies, it is quite important to have as wide range of spectrum as possible (it is particularly important to assess the reliability of tree

rings for a frequency range from centennial to one thousand years). While the ice-core spectrum is provided for a few years to 500 years, the range is quite limited for tree rings and corals. Particularly, there should be tree-ring records available to discuss P > 100 years.

Authors: The reviewer is correct that there are very long (1000 year +) tree-ring records available in the PAGES data. However, they are the minority (see PAGES Consortium 2017, Fig. 1c), and most records are less than 500 years old. The clustering method of SNR estimation is limited by the shortest chronology in a cluster, and the lowest two frequencies are removed to reduce bias, such that there are only a few clusters that reach multi-centennial timescales. These are now included in Appendix C (Fig C1 a,b).

The authors mention on Line 234 (Appendix B) that they have 186 clusters of data. I assume that the analyses were conducted independently, as the authors also mention that they analyzed "co-located records" on Line 91. Further details are needed on how Figure 3 (a) and (d) were derived from the 186 clusters.

Authors: We have changed 'co-located' to 'nearby' to reduce confusion (co-located implies pairs), and included additional details on the clustering method in Appendix B. (Line 114 and 310-326)

**2. Discussion**

Sections 2 and 3 contain only a small portion of the authors' own results, while a significant amount is dedicated to reviewing previous work, and it is not clear what new insights this study provides. The text may be reorganized to emphasize the author's findings.

Line 170 "if the power of the noise rises more steeply than the signal": For the comprehensive understanding of the behavior of the noise and signals, it is desired to show signal PSD as done in Muench and Laepple (2018), and even the original PSD of the data, in addition to the noise PSD and SNR, either in the main text or in the appendix.

Authors: We added the original (proxy) and corrected (i.e. signal) PSD in the Appendix (Fig C1), with an explanation of the figure (Appendix C). We opted not to add this to the main text as our focus is on the noise component rather than the climate signal, which would require outside comparisons with other data types.

Line 174 "The color of the noise thus determines at which timescales a robust climate signal can be reconstructed": Shouldn't the amplitude of the noise be much more critical than its color?

Authors: We acknowledged that both are important. We have added an additional sentence "The color of the noise thus determines at which timescales a robust climate signal can be reconstructed because it introduces a frequency-dependence to the signal-to-noise ratio." (Line 252-253)

Further details are needed for the discussions regarding the cause of the noise spectrum characteristics. For example, the authors seem to suggest that large seasonal variability contributes to the white-noise characteristics of ice-core records; however, the noise PSD is relatively high across the whole range from 10 years to 100 years (Figure 3c), although the authors mention that they "fail in many regions to reconstruct interannual to decadal changes". The frequency range should be specified when proposing the hypothesis for the cause of the noise.

Authors: For tree rings and corals, we expanded substantially on the mechanisms that contribute to the noise and on what timescale they operate (i.e. biological memory, detrending effects, and non-linear growth responses) (Lines 154-177). For ice cores we clarify the mechanisms underlying the white noise level in ice cores as being precipitation intermittency and post-depositional redistribution, which break up the signal of the large seasonal cycle and are redistributed as white noise across lower frequencies (191-193)

Lines 114-117: The relatively lower SNR for corals appears on the timescale of 10-30 years, and it is not a "slow" change. I feel that the proposed reasons suggested here are not for this short timescale (here I assume the lifespan of corals are much longer, as the lengths of the data are not indicated in Appendix B). The proposed causes of the noise should specifically correspond to the frequency range under discussion.

Authors: In the case of corals we are not thinking about ageing/ontogenic effects as in trees, rather shifts in biology in response to stress events. We have added a new paragraph in the discussion (Lines 172-177) exploring the possible reasons for non-linear temperature responses in corals. We have also added the density plot in Appendix B showing the replication at different timescales (Fig C1.d). The longest coral timeseries are over 100 years long, but on average the length of the overlapping period on which the SNR can be calculated from the clustered records is around 40 years.

**References**

PAGES2k Consortium et al. A global multiproxy database for temperature reconstructions of the Common Era. Scientific Data **4**, 170088 (2017). <a href="https://doi.org/10.1038/sdata.2017.88">https://doi.org/10.1038/sdata.2017.88</a>

---

## Author Response (AR2)

**General comments**

Authors: We thank the two reviewers for taking the time to re-read and comment on our project, and we are glad to learn that the consensus is that the project has been improved in their estimation. We have adopted the suggestions made, and attempted to clarify issues that remained unclear. We agree with both reviewers, and with the editor, that this project would be best classified as a review paper, rather than a research article.

**Reviewer #1**

In my view, the manuscript has been improved through revision, but I still have some recommendations, primarily regarding phrasing and language use. Also, after reading this revised version, I believe that the manuscript has not been correctly classified, as it seems more like a review article rather than a research article to me. This point may justify some of the critical comments of the other reviewers regarding the lack of an explicit research question

**Main points**

1) The manuscript presents a review of the types of noise found in different classes of proxy records. As such, I found it informative, as it summarises information scattered throughout the paleoclimate literature on the nature of proxy noise and offers some interesting insights into its interpretation. However, the manuscript does not clearly address a stated research gap, so I would not classify it as a research paper.

Authors: We would agree with this assessment, and would be very happy if the paper were published as a review article in CP.

2) The language used, mainly in the introduction, is too often unspecific, too ambiguous and thus more challenging to grasp than it should. I have included some examples below, but the authors may want to review the introduction again with this comment in mind.

**Particular points:**

3) 'For example, "white noise" equally affects all timescales, where "red noise" dominates only on long timescales, similar to longwave red light. ..'

This paragraph and another one of the same nature later in the manuscript, explaining the meaning of the red spectrum or blue spectrum, are, in my view, not necessary for the readership of Climate of the Past.

Authors: We appreciate this comment, but we are here responding to a comment by the editor, who requested a more general overview of colored noise, and color in time series more generally, so we have determined that –if classified as a review paper – it is appropriate to leave in these more general statements.

2) Here, we provide concrete definitions of types of timescales-dependent errors and review methods for estimating these errors in different types of proxy data

I would not agree with this sentence. Do the authors mean that the definitions 'red,blue,or white' need to be defined again, or do they mean something else? If they mean something else, I did not see those definitions later in the manuscript. It can be confusing for some readers.

Authors: Here are referring to the distinction we draw between noise which we consider to be generally stochastic and smoothing, which produces error but through a reduction in variance. We have edited this phrasing to be more general (Line 14-16).

3) 'We then synthesize the results of several studies that use a common empirical approach for estimating the noise spectrum in ice core'

This sentence is to me quite unclear, perhaps due to the use of the word 'synthesise'. In my understanding, this word means to produce or to combine, not to summarise (?). I understood the sentence as if the authors were to recreate the results of previous studies.

Authors: This is correct – we are taking single results from multiple published students and reproducing them.

4) Modification of proxy records can result in the addition of variance

What does 'modification' mean here? does it mean to change a posteriori the numerical values of the proxy record or does it refer to natural process that modify the climate signal in the proxy? After reading this paragraph several times, I was still left in doubt.

Authors: It could be either, an 'a posteriori' effect resulting from observational or measurement error, or from a natural process might be redistribution of sediments or snow, or smoothing due to previous year's effects over growth, for example. We have edited this sentence to improve clarity (Line 35).

5 ) 'whereas the loss of variance through smoothing also constitutes error, but not technically noise.'

Again, I had to read this sentence several times, unsuccessfully. Does it mean that smoothing may introduce an additional uncertainty or bias (error?)? I would not generally agree with it. Continuous physical smoothing, in theory, reduces the time resolution, but it does not introduce per se uncertainty or bias.

Authors: We would agree that smoothing can reduce time resolution, although we would also argue that smoothing introduces uncertainty. For example, in tree rings, biological smoothing doesn't reduce time-resolution since tree rings can still be precisely dated. However, it does introduce uncertainty, for example in estimates of past climate anomalies which may appear

dampened or extended due to this memory. We have edited this line to include a definition of error as any discrepancy between true and reconstructed climate at a given timescale (Line 41).

6) 'By contrast, noise is typically independent,...

Independent of what?

Authors: Independent between records – clarification made (Line 58)

7) 'The timescale-dependent variations of a time series can be analyzed in the spectral domain and referred to using colors by loose analogy to the frequency spectrum of light ...'

Again, the authors may want to reconsider whether this explanation is really necessary

Authors: Here, we were responding to the previous comment by the editor requesting an overview of color in time series more generally so we have opted to leave this section.

**8) Section 2**

This section starts with the sentence 'Our synthesis demonstrates that tree rings and corals ...' but the reader will wonder where these results come from . How was the noise estimated? Or do the authors rely on noise estimations from the original publications? The reader needs some information here.

Authors: We have added the citations to the original studies at the end of this phrase and the one below on ice cores (Lines 173, 223)

9) 'the proxy signal which, in theory, can be accounted for using deterministic modeling..'

accounted for by using... is perhaps clearer for non-native speakers, at least for me.

Authors: Edit made (Line 252)

**Reviewer #2**

The manuscript in its new version was substantially improved, including additional studies and according references, rendering the style of the manuscript into a comprehensive review paper rather than a summary of individual studies published earlier by the authors. The new version includes also an extension of the main chapters on proxy noise and its implications, and additional chapters on signal and signal-to-noise estimates and simulated effects of time uncertainty within the appendix. It is also appreciated that the comments raised in the first review round were comprehensively addressed, putting the main criticisms into context.

Being a review and synthesis paper on this specific topic, I think in its present form the manuscript is now suited for publication in Climate of the Past.

Specific minor comments:

Headings for sections 2 and 3:

I suggest to extend the headings being a bit more specific. e.g. The colours of proxy noise in different archives and Implications of proxy noise on time scales of climate signals or similar.

Authors: Change made (Headings sections 2&3)

Figure 2:

Is the right column intended to be on a log-log scale? Then it might be advisable to adapt the ticks for the axes resembling a log-log plotting structure with irregular spacing of the x- and y-ticks (cf. Fig. 3).

Authors: It is a log-log scale, we have modified the tick marks as suggested, and note that it is also mentioned in the caption that the noise spectra are visualized on a log-log plot (Fig 2).

Code Availability:

The authors might add a link to the respective software repositories on Zenodo to easier and more efficiently access the code.

Authors: We have added the appropriate software citations to the reference list (Reference section Lines 720)

---

## Author Response (AR3)

Response to Editor Comment: "1. The ROR database lists the institution of the corresponding author but with a different city than given in the manuscript`s affiliation. Please clarify whether the ROR in the system "Alfred-Wegener-Institut Helmholtz-Zentrum für Polar- und Meeresforschung (Bremerhaven, Germany)" is still correct. 2. Your reference list includes works "under revision". Such works can be cited upon submission if being available to the reviewers. They should not be cited in the final, accepted manuscript, unless published, accepted for publication, or available as preprint with a DOI."

**Author response:**

- 1) The author affiliation "Alfred-Wegener-Institut Helmholtz-Zentrum für Polar- und Meeresforschung, Research Unit Potsdam, Potsdam, Germany" is correct as written on the manuscript title page.
- 2) We have corrected all references to Dolman *et al.* (under revision) to Dolman *et al.* (2025). The preprint is included in reference list, and it is our hope that the paper will be accepted soon, as it has already undergone major revisions.